# Role of Ultrasound in Evaluating Ligament Injuries Around the Ankle: A Narrative Review

**DOI:** 10.3390/diagnostics15030392

**Published:** 2025-02-06

**Authors:** Soichi Hattori, Rachit Saggar, Joseph Mullen, Abdulganeey Olawin, Eva Heidinger, Warren Austin, Akeem Williams, Glenn Reeves, MaCalus Vinson Hogan

**Affiliations:** Foot and Ankle Injury Research (FAIR), Division of Foot and Ankle, Department of Orthopaedic Surgery, University of Pittsburgh Medical Center, Pittsburgh, PA 15219, USA; saggarr2@upmc.edu (R.S.); mullenjp3@upmc.edu (J.M.); olawinao@upmc.edu (A.O.); heidingerem@upmc.edu (E.H.); austinwp@upmc.edu (W.A.); williams.akeem@medstudent.pitt.edu (A.W.); greeves18@email.mmc.edu (G.R.); hoganmv@upmc.edu (M.V.H.)

**Keywords:** musculoskeletal ultrasound, anterior talofibular ligament, calcaneofibular ligament, syndesmosis, anterior inferior tibiofibular ligament, deltoid ligament, diagnostic ultrasound

## Abstract

Ultrasound has emerged as a valuable imaging modality for evaluating ligamentous injuries around the ankle joint, offering several advantages over traditional imaging techniques. It is more cost-effective and widely available than MRI, and it avoids the ionizing radiation exposure associated with X-rays, making it a safer option, particularly for pediatric and adolescent populations. In cases of inversion ankle sprains, ultrasound allows for more accurate assessment of the anterior talofibular ligament (ATFL) and calcaneofibular ligament (CFL) compared to X-rays and manual examination and yields diagnostic results comparable to MRI. For high ankle sprains involving syndesmosis injuries, ultrasound—especially stress ultrasound—has shown high diagnostic accuracy. Additionally, ultrasound evaluation of the deltoid ligament (DL) in cases of ankle fractures can aid surgeons in determining the need for ligament repair in conjunction with fracture management. This review explores recent developments in ultrasound imaging of the lateral, medial, and syndesmotic ligaments of the ankle joint and discusses its potential applications for evaluating the spring and posterior ligaments. The review provides a comprehensive overview of the ever-expanding role of ultrasound in the management of ankle ligamentous injuries.

## 1. Introduction

Musculoskeletal (MSK) ultrasound is a highly versatile imaging modality widely used for diagnostic [1,2,3] and therapeutic purposes [4,5] in a variety of clinical settings. Compared to other imaging modalities, such as magnetic resonance imaging (MRI), MSK ultrasound provides several key advantages. It offers exceptional spatial resolution, enabling detailed visualization of superficial structures such as tendons, ligaments, muscles, and nerves. Unlike imaging methods such as X-rays or computed tomography (CT) that rely on ionizing radiation, ultrasound is entirely radiation-free, making it a safer option for repeat assessments [6]. This feature makes ultrasound imaging especially useful for pediatric or pregnant patients.

One of the most notable benefits of MSK ultrasound is its real-time imaging capability that allows for dynamic evaluation of joint movements, tendon gliding, and other functional assessments that are impossible with static imaging techniques like MRI. The portability of ultrasound devices makes it ideal for bedside or outpatient evaluations and makes it a more accessible tool in diverse clinical environments, such as rural areas. Furthermore, MSK ultrasound is cost-effective, reducing the financial burden for both healthcare systems and patients. These advantages, combined with its non-invasive nature and ability to guide therapeutic interventions (such as injections), highlight the critical role of ultrasound in routine MSK care [7].

Ankle injuries encompass both sprains and fractures. Ankle sprains are among the most common sports-related injuries, while ankle fractures rank as one of the most frequent injuries to the lower extremity. MSK ultrasound is a particularly powerful diagnostic tool in ankle injuries, where it offers several advantages over more traditional imaging modalities. It provides real-time visualization and insight into the integrity of structures such as the anterior talofibular ligament (ATFL), calcaneofibular ligament (CFL), syndesmosis including the anterior tibiofibular ligament (AiTFL), and the deltoid ligament (DL). Ultrasound is particularly useful for diagnosing ankle sprains, especially inversion injuries (to the ATFL and CFL) as it provides a more accurate assessment than X-rays or manual examination. Furthermore, in cases of high ankle sprain and ankle fractures, ultrasound demonstrates superior diagnostic accuracy. It also can help surgeons assess the need for ligament repair in conjunction with fracture management.

The application of MSK ultrasound has undergone significant progression and evolution over the past 30 years, transitioning from experimental beginnings to becoming an integral tool in clinical practice. In the 1990s, foundational cadaveric studies were pivotal in demonstrating the potential utility of MSK ultrasound in the evaluation of ankle ligament integrity. These studies provided the first detailed sonographic insights into specific ligaments, starting with the ATFL, CFL, and DL in 1993, followed by the AiTFL in 1996 [8,9]. These investigations not only confirmed the feasibility of ultrasound for visualizing these structures but also set the stage for expanding its application into clinical scenarios.

This transition from preliminary research to clinical integration emerged gradually over the subsequent decades. During the late 1990s and early 2000s, exploratory studies focused on translating cadaveric findings into live patient settings. Early clinical studies often involved smaller sample sizes and primarily aimed to validate the imaging characteristics of various ligaments. By the late 2010s, high-volume prospective studies gained momentum. This period marked a key shift toward the evidence-based refinement of MSK ultrasound diagnostic capabilities and techniques.

Advancements in MSK ultrasound technology over the past decade, such as higher-resolution capabilities, have helped ultrasound become a standard imaging modality in orthopedic settings. High-level systematic reviews and meta-analyses published during the past decade have provided strong clinical evidence supporting its diagnostic accuracy and clinical utility. For instance, systematic reviews and meta-analyses addressing the ATFL in 2016, the CFL in 2018, and the DL in 2022 have reinforced the utility and accuracy of MSK ultrasound in evaluating these ligaments [10,11,12,13]. This historical progression underscores how MSK ultrasound has transformed from a research tool into a cornerstone of musculoskeletal imaging that is continually evolving to meet the needs of modern clinical practice.

Despite its diagnostic advantages, MSK ultrasound remains underutilized in clinical practice. This is evident across the world, where limited training during orthopedic residency programs and the lack of experienced operators restricts its widespread adoption. Barriers to the adoption of MSK ultrasound include the need for specialized training, limited exposure during medical education, and reliance on more established imaging modalities like MRI and X-rays.

This paper aims to provide a comprehensive review of the role of MSK ultrasound in diagnosing and treating ligament injuries around the ankle. It focuses on key ligaments, including the ATFL, CFL, syndesmosis including the AiTFL, DL, as well as other structures such as posterior ligaments and the spring ligament. By integrating the latest research findings with practical insights, this review aims to equip clinicians and researchers with a deeper understanding of the expanding utility of MSK ultrasound in the diagnosis and management of ligamentous injuries around the ankle.

## 2. Materials and Methods

### 2.1. Literature Search

This review evaluated the diagnostic utility of ultrasound for ankle ligament injuries. We included peer-reviewed studies in English, focusing on healthy human subjects or subjects with suspected or confirmed ligament injuries. Eligible designs included prospective and retrospective studies, systematic reviews, meta-analyses, cadaveric studies, and case series. Exclusion criteria included Level 5 evidence studies, such as case reports, pictorial essays, narrative reviews, and expert opinions. Additionally, abstracts, animal studies, and studies lacking clear methodologies, diagnostic criteria, or quantitative data were excluded.

We conducted a comprehensive literature search using PubMed and EMBASE databases from inception to February 2024. The search strategy included combinations of ankle ligament terminology (“anterior talofibular ligament”, “calcaneofibular ligament”, “posterior talofibular ligament”, “deltoid ligament”, “syndesmosis”, “tibiofibular ligament”, “spring ligament”) paired with imaging terms (“ultrasound”, “ultrasonography”, “sonography”, “USG”) in title/abstract fields. Boolean operators (AND, OR) were used to combine search terms appropriately. Additional relevant articles were identified through the manual reference-checking of included studies.

### 2.2. Classification of Studies

The initial search yielded 698 articles. After removing duplicates, 316 articles remained for screening. Two independent reviewers screened titles and abstracts, followed by full-text review of potentially eligible studies. Disagreements were resolved through discussion or by consulting a third reviewer. Following title and abstract review, 151 articles were selected for full-text assessment. Of these 152, 99 studies met the inclusion criteria (Figure 1).

Data extracted from included studies comprised year of publication, study design, ultrasound technique, anatomical structure in focus, reference standards used, types of injury, sample characteristics, and key findings.

### 2.3. Level of Evidence Assessment

Studies were categorized according to Journal of Bone and Joint Surgery guidelines for level of evidence [14]. Level I evidence included high-quality randomized controlled trials and systematic reviews/meta-analyses with homogeneous Level 1 studies. Level 2 evidence comprised lesser-quality randomized controlled trials, prospective comparative studies, and systematic reviews of Level 2 studies. Level 3 evidence included case–control studies and retrospective comparative studies. Level 4 evidence consisted of case series with no controls, and Level 5 included expert opinion and case reports. For diagnostic studies specifically, level of evidence was determined based on study design, consecutive injured patient enrollment, consistent reference standard application, and the adequate blinding of observers. Studies above Level 3 evidence had to report ultrasound diagnostic accuracy metrics with comparisons to other imaging modalities or operative findings. Systematic reviews and meta-analyses were classified according to the level of evidence of their included studies.

## 3. Results

### 3.1. Anterior Talofibular Ligament (ATFL)

Ankle sprains are among the most common sports-related injuries, accounting for approximately 30% of all sports injuries. The anterior talofibular ligament (ATFL), one of the lateral ligaments of the ankle, is the most frequently injured ligament in ankle sprains. Due to its superficial location and high injury prevalence, the ATFL has been extensively studied using ultrasound. With ultrasound, the entire portion of the ligament can be visualized in a longitudinal view. A normal ATFL can be seen as a band-like fibrillar hyperechoic structure [15,16]. Hypoechoic areas typically indicate the presence of abnormality in the ligament.

#### 3.1.1. Level 4 Evidence Studies

This review included 52 papers classified as Level 4 evidence, focusing on the use of ultrasound to assess an intact or injured ATFL. Among these, 5 were case series [17,18,19,20,21], 9 were cadaveric studies [9,22,23,24,25,26,27,28], and 38 lacked reliable reference standard imaging modalities or involved healthy subjects [15,29,30,31,32,33,34,35,36,37,38,39,40,41,42,43,44,45,46,47,48,49,50,51,52,53,54,55,56,57,58,59,60,61,62]. There was a consensus amongst the authors that stress ultrasound can accurately evaluate AFTL injuries by providing insights into the differences in the AFTL length, thickness, and laxity when compared to a healthy AFTL [35,43,45,46,49]. Some studies demonstrated an AFTL that was thicker and longer in patients with injury [33,35,41,45,46,49,61].

#### 3.1.2. Clinical Studies Above Evidence Level 3

Twenty-two papers describing the use of ultrasound for the ATFL had a level of evidence (LOE) that was above three. Six of these papers were retrospective studies (LOE 3) [15,29,30,31,32,33,34,35,36,37,38,39,40,41,42,43,44,45,46,47,48,49,50,51,52,53,54,55,56,57,58,59,60,61,62], and nine were prospective studies (LOE 2). The remaining seven papers were systematic reviews and/or meta-analyses. The papers above LOE 3 involved patients with injured ankles with the diagnostic reference standards being MRI or surgical finding including open and arthroscopy or arthrography. In some cases, a combination of these was used as the reference standard (Table 1).

Overall, these studies demonstrated that ultrasound has a high diagnostic accuracy in detecting ATFL injuries. Recent systematic reviews and meta-analysis showed that the pooled sensitivity and specificity of ultrasound to ATFL injuries was greater than that of MRI [63].

Nine of these papers evaluated acute ATFL injuries, eight evaluated chronic ATFL injuries, three were a mix of both, and two systematic review and meta-analyses did not indicate whether there were acute or chronic ATFL injuries. The diagnostic performance of ultrasound for chronic ATFL injuries demonstrated consistency, with sensitivity ranging from 84% to 100% and specificity ranging from 91% to 100%. In contrast, for acute ATFL injuries, the sensitivity was more variable, ranging from 57% to 97%, while specificity also fluctuated between 64% and 100%. The variability observed in the acute setting can be attributed to the limitations of aggressive dynamic ultrasound examinations, primarily due to the significant pain associated with acute injuries. Furthermore, swelling and hematoma within the subcutaneous tissue may obscure the ligament and displace it to a deeper location, further reducing the accuracy of ultrasound visualization. In contrast, stress ultrasound is more feasible in sub-acute and chronic settings. Its utility in evaluating ATFL instability has been well-documented [22,61,64,65]. This technique enhances the reliability and accuracy of ATFL evaluations, particularly in cases of chronic ankle instability.

Ultrasound has been extensively studied for ATFL evaluation, showing high diagnostic accuracy, particularly in chronic cases. In acute injuries, diagnostic accuracy is slightly lower, likely due to the impracticality of performing aggressive dynamic examinations.

### 3.2. Calcaneofibular Ligament (CFL)

The calcaneofibular ligament (CFL), along with the ATFL and the posterior talofibular ligament, forms the lateral ligament complex of the ankle joint, wrapping around the peroneal tendons and spanning the tibiotalar and subtalar joints. As the second most commonly injured ligament in ankle sprains, CFL involvement classifies the injury as grade 3 in Beynnon’s classification [66]. While ultrasound is often used to evaluate the CFL alongside the ATFL in ankle sprains, its anatomical position beneath the peroneal tendons makes it challenging to visualize the entire ligament in a long-axis view in particular.

#### 3.2.1. Level 4 Evidence Studies

For CFL diagnostics, 15 papers were classified as Level 4 evidence, studying the use of ultrasound to assess intact or injured CFL. Among these, one was a case series [19], seven were cadaveric studies [8,9,23,26,27,67], and eight lacked reliable reference standard imaging modalities or involved healthy subjects [17,24,53,54,56].

#### 3.2.2. Clinical Studies Above Evidence Level 3

A total of 13 diagnostic studies included in the analysis met or surpassed evidence LOE 3, with 3 systematic reviews and meta-analyses (Table 2). No RCTs were identified that evaluated ultrasound’s efficacy in CFL injuries. These systematic reviews and meta-analyses focused primarily on the diagnostic utility of ultrasound for lateral ankle ligament injuries, evaluating the CFL in addition to the ATFL. The remaining 10 studies were classified as LOE 2 or 3, meeting criteria for rigorous methodology, defined as patient populations with confirmed ankle injuries and reference standards including MRI, surgery, or arthrography.

Most of these studies (12 of 13) validated ultrasound’s diagnostic legitimacy for CFL injuries [12,15,65,68,69,70,71,72,73,74]. These investigations evaluated the CFL in the context of other ligaments in 12 of the studies (12 ATFL, 4 AITFL, 2 Deltoid, 2 PTFL injuries, and 1 Achilles Tendon injury).

**Table 1 diagnostics-15-00392-t001:** Clinical studies of anterior talofibular ligament (ATFL) ultrasound evaluation above Evidence Level 3.

Author	Year	LOE	Sample Size	Design	Injury Types	Reference Standard	Key Findings
Kocsis [68]	2024	2	8 studies with 434 patients	SR and MA	Acute	Not specified	97% sensitivity, 93% specificity
Colò [63]	2023	2	4 studies with 207 for US	SR and MA	Not specified	Not specified	Greater sensitivity of 97% for US compared to MRI
Cao [75]	2023	2	25 studies with 1409	SR and MA	Acute Chronic	Not specified	88.6% sensitivity, 90.3% specificity for acute, 98.7%, 94.0% for chronic injuries
Ergün [74]	2022	2	30 patients	Prospective diagnostic	Acute	MRI	95.4% sensitivity, 75% specificity
Hosseinian [73]	2022	3	105 patients	Cross sectional	Acute Chronic	MRI	100% sensitivity and specificity for intact ATFL
Baltes [76]	2021	2	117 patients	Prospective diagnostic	Acute	MRI	87% sensitivity, 69% specificity
Esmailian [70]	2021	3	31 patients	Cross sectional	Acute	MRI	66.7% sensitivity, 92.9% specificity
Lee [77]	2020	2	10 studies with 443	SR and MA	Acute Chronic	Surgery	0.95 accuracy (0.92 acute and 0.96 chronic)
Seok [65]	2020	2	10 studies with 380	SR and MA	Not specified	Not specified	0.99 sensitivity, 0.92 specificity
Cao [12]	2018	2	15 studies with 695	SR and MA	Chronic	Not specified	99% sensitivity, 91% specificity
Elkaïm [78]	2018	2	63 patients for US study	Prospective diagnostic	Chronic	Arthroscopy	76.2% agreement
Jones [79]	2018	2	7 pediatric patients	Prospective diagnostic	Acute	MRI	Good specificity (86%), Poor sensitivity (57%)
Lee [69]	2017	2	85 patients	Prospective diagnostic	Chronic	MRI	98.5–100% sensitivity, 95.0% specificity
Radwan [10]	2016	2	6 studies with 431	SR	Chronic	Not specified	84.6–100% sensitivity, 90.9–100% specificity
Cho [64]	2016	3	28 patients	Case–control	Chronic	Arthroscopy	100% agreement
Cheng [16]	2014	2	120 patients	Prospective diagnostic	Chronic	Surgery	98.9% sensitivity, 96.2% specificity, 84.2% accuracy
Gün [80]	2013	2	65 patients	Prospective diagnostic	Acute	MRI	93.8% sensitivity, 100% specificity
Hua [81]	2012	2	83 patients	Prospective diagnostic	Chronic	Arthroscopy	97.7% sensitivity, 92.3% specificity
Guillodo [82]	2010	2	56 patients	Prospective diagnostic	Chronic	Arthrography CT	Substantial agreement
Milz [83]	1998	2	20 patients	Prospective diagnostic	Acute	MRI	92% sensitivity, 83% specificity
Van Dijk [84]	1996	2	74 patients for US study	Prospective diagnostic	Acute	Arthrography and Surgery	92% sensitivity, 64% specificity
Friedrich [71]	1993	2	105 patients	Prospective diagnostic	Acute	Surgery	100% agreement

SR, systematic review; MA, meta-analysis; US, ultrasound.

**Table 2 diagnostics-15-00392-t002:** Clinical studies of calcaneofibular ligament (CFL) ultrasound evaluation above Evidence Level 3.

Author	Year	LOE	Design	Sample Size (Only CFL if Applicable)	Injury Types	Reference Standard	Key Findings
Kocsis [68]	2024	2	SR and MA	6 studies with 346 patients	Acute	Not specified	81% sensitivity, 92% specificity
Ergün [74]	2022	2	Prospective diagnostic	30 patients	Acute	MRI	76.9% sensitivity, 100% specificity
Hosseinian [73]	2022	3	Cross sectional	105 patients	Acute Chronic	MRI	93% sensitivity for normal CFL, highly specific in detecting torn ligament
Esmailian [70]	2021	3	Cross sectional	31 patients	Acute	MRI	100% sensitivity, 93.1% specificity
Baltes [76]	2021	2	Prospective diagnostic	117 patients	Acute	MRI	49% sensitivity, 91% specificity
Alvarez [72]	2020	3	Retrospective diagnostic	21 patients	Chronic	MRI	90% sensitivity, 100% specificity,
Seok [65]	2020	2	SR and MA	Not specified	Not specified	Not specified	0.95 sensitivity, 0.99 specificity
Cao [12]	2018	2	SR and MA	1 study with 120 patients	Chronic	Not specified	93.8% sensitivity, 90.9% specificity
Lee [69]	2017	2	Prospective diagnostic	85 patients	Chronic	MRI	96.4–100% sensitivity, 96.5–100% specificity
Cheng [16]	2014	2	Prospective diagnostic	120 patients	Chronic	Surgery	93.8% sensitivity, 90.9% specificity, and 83.3% accuracy
Milz [83]	1998	2	Prospective diagnostic	20 patients	Acute	MRI	100% sensitivity, 100% specificity
Friedrich [71]	1993	3	Prospective diagnostic	105 patients	Acute	Surgery	Agreed with operative findings in 92%

SR, systematic review; MA, meta-analysis.

Of these studies, seven targeted patients with acute CFL injuries, and four focused on chronic ankle instability. The sensitivity of ultrasound evaluation for acute CFL injuries is highly variable, ranging from 49% to 100%, while the specificity remains consistently high, at 89% to 100% [68,70,71,73,74,76,83]. In contrast, for chronic CFL injuries, both sensitivity and specificity are consistently high, ranging from 90% to 96.4% and 90.9% to 100%, respectively [12,16,69,72,73,74]. The variability in sensitivity for acute CFL injuries can be attributed to the ligament’s anatomical position beneath the peroneal tendons, which allows only the calcaneal side to be visualized in the acute setting. As demonstrated in a cadaveric study [67], maximal dorsiflexion of the ankle is necessary to visualize the fibular side of the CFL; however, this maneuver is often unachievable in acute injuries due to pain. Furthermore, swelling and hematoma in the subcutaneous tissue can displace the ligament to a deeper location, further reducing the accuracy of ultrasound visualization.

Ultrasound evaluation for CFL injury demonstrated high specificity for acute CFL injuries and consistent diagnostic accuracy in chronic injuries. Its sensitivity in acute cases was slightly more variable, most likely due to factors like swelling, hematoma, and the inability to perform maximal dorsiflexion in acute settings.

### 3.3. Syndesmosis Including the Anterior Inferior Tibiofibular Ligament (AiTFL)

The syndesmosis, comprising the anterior inferior tibiofibular ligament (AiTFL), posterior inferior tibiofibular ligament, interosseous ligament, and interosseous membrane, plays a critical role in maintaining ankle joint stability. Syndesmotic injuries account for up to 18% of ankle sprains in the general population and up to 74% of ankle sprains in athletes [85], yet their diagnosis remains challenging due to subtle clinical presentations and the complex biomechanics of the ankle joint. While MRI and CT have traditionally been considered diagnostic mainstays, ultrasound offers unique advantages, including real-time, dynamic assessment capabilities that enable the evaluation of ligament integrity and joint function under physiological stress. Previous ultrasound studies on syndesmosis injuries have primarily focused on evaluating the AiTFL, as it is the most superficially located ligament among the syndesmotic ligaments. Furthermore, many of these studies have investigated the dynamic assessment of syndesmotic instability.

#### 3.3.1. Level 4 Evidence Studies

This review included seven papers classified as Level 4 evidence, focusing on the use of ultrasound to assess an intact or injured syndesmosis including the AiTFL. These studies primarily focused on dynamic ultrasound evaluation of syndesmotic instability, as it is a critical factor for determining surgical indications in high ankle sprains and certain ankle fractures. Mei-Den showed that healthy adults exhibit baseline tibiofibular clear space values of 3.78 mm in neutral positioning and 4.08 mm in external rotation, reflecting a difference of 0.3 mm [86]. Fisher demonstrated in their cadaveric study that ultrasound detection of tibiofibular clear space widens from 4.5 mm to 6.0 mm (a 1.5 mm difference) in Stage 1 Lauge-Hansen Supination-External Rotation (SER) injuries [87].

The collective of the literature highlighted several clinical insights. Dynamic ultrasound assessment should position the transducer 1 cm proximal to the tibiotalar joint, with external rotation stress consistently revealing more pathology than neutral positioning. Side-to-side comparisons offer greater reliability than absolute measurements, and weight-bearing evaluations frequently uncover instability that is not apparent in traditional positioning [50,53,88,89,90,91,92].

#### 3.3.2. Clinical Studies Above Evidence Level 3

Seven papers describing the use of ultrasound for diagnosing AiTFL injuries had a LOE above 3. Seven were prospective studies with an LOE of 2, while one was a retrospective study (LOE 3). These studies included patients with injured ankles, using MRI as the diagnostic reference standard (Table 3). Overall, the findings support the role of ultrasound in diagnosing syndesmotic injuries.

Of the seven papers, three focused on the static ultrasound evaluation of the AiTFL [69,73,74], while three assessed the diagnostic value of the dynamic ultrasound examination for syndesmotic injuries [76,86,93]. The focus of one paper was unknown [73]. Most studies on static evaluation were conducted in the setting of acute syndesmotic injuries. Sensitivity and specificity ranged from 66% to 100% and 75% to 100%, respectively, using MRI as the reference standard. These values were variable, similar to our findings in ultrasound evaluations of ATFL and CFL injuries in acute-care settings. The variability is likely attributable to factors such as swelling, hematoma, and the inability to perform dynamic examinations due to pain in acute injuries.

Three studies specifically involved dynamic ultrasound for syndesmotic injuries. One study, conducted in the acute-care setting, reported sensitivity and specificity inferior to those of static evaluations [76], likely due to the challenges associated with acute injury conditions. The other two studies, conducted in sub-acute syndesmotic injury settings, reported sensitivities of 79% and 89% and a specificity of 100% for detecting AiTFL injuries. These studies demonstrated their protocols that measured tibiofibular clear space under stress conditions [86,93]. However, a limitation of this approach can be the small cut-off distance (less than 1 mm) between stressed and unstressed states, which may be challenging to detect for inexperienced ultrasound operators.

In summary, dynamic ultrasound evaluations of the syndesmosis, together with the visualization of the AiTFL, demonstrated high accuracy in sub-acute settings, whereas mild variability in sensitivity and specificity was observed in acute injuries, which could be influenced by the impracticality of aggressive dynamic ultrasound evaluation due to pain.

### 3.4. Deltoid Ligament (DL)

The deltoid ligament (DL) is a strong, triangular structure on the medial side of the ankle, extending from the medial malleolus to various foot bones. It consists of superficial and deep layers: the superficial layer attaches to the navicular bone, spring ligament, and sustentaculum tali, while the deep layer anchors the talus through the anterior and posterior tibiotalar ligaments. This ligament provides medial stability by preventing excessive eversion and external rotation of the ankle and stabilizing the subtalar and tibiotalar joints during weight-bearing activities. Injuries typically result from forced eversion, often in conjunction with lateral malleolar fractures. They can also occur due to inversion ankle sprains especially in cases of chronic ankle instability. Ultrasound evaluation of the DL is commonly performed in two contexts: ankle fractures and ankle sprains.

#### 3.4.1. Level 4 Evidence Studies

For DL diagnostics, two papers were classified as Level 4 evidence. One was a cadaveric study [8], and the other was a case series of ultrasound evaluations of DL injuries [94].

#### 3.4.2. Clinical Studies Above Evidence Level 3

Eight papers describing the use of ultrasound for diagnosing DL injuries had a LOE above 3, including two systematic reviews and meta-analyses that focused on ankle fractures (Table 4). Five were prospective studies with an LOE of 2, while one was a retrospective study (LOE 3). These studies evaluated patients with ankle fractures or ankle sprains, using MRI, surgical findings (arthroscopic or open surgery), arthrography, or gravitational stress radiography as the diagnostic reference standards.

Overall, these studies demonstrated that ultrasound has high diagnostic accuracy for detecting DL injuries. Systematic reviews and meta-analyses reported that the pooled sensitivity and specificity of ultrasound were superior to those of other imaging modalities, such as MRI [11,13]. However, for complex or deep DL injuries, MRI showed slightly higher sensitivity and specificity [13].

Of the six papers, four focused on ultrasound evaluation in ankle fractures [86,94,95,96], while two assessed its diagnostic accuracy in ankle sprains [68,72]. In the context of ankle fractures, six studies consistently reported high diagnostic accuracy for ultrasound in detecting complete tears of the deep DL, which can lead to medial ankle instability, with sensitivity and specificity ranging from 94.7% to 100% and 90% to 100%, respectively. Although one study reported a specificity of 66.4% for detecting deep DL injuries, it demonstrated 100% specificity for detecting complete deep DL tears [96]. Two studies evaluated DL injuries in the context of ankle sprains alongside other ligaments including the ATFL. These studies reported sensitivities of 90% and 100% and specificities of 78% and 97%, both of which indicate moderately high diagnostic accuracy [68,73].

This review on the DL injuries demonstrated that ultrasound is highly effective in diagnosing both fracture-related and sprain-related DL injuries. Six studies, including systematic reviews, highlighted its high diagnostic accuracy, particularly for detecting complete DL tears associated with fibular fractures, with consistently high sensitivity and specificity.

### 3.5. Other Peri-Ankle Ligaments

One study investigated the feasibility of visualizing the posterior ankle ligaments, which are located deep within the ankle and are challenging to assess, using healthy individuals [89]. While the study demonstrated the feasibility of using ultrasound to visualize these ligaments, no clinical studies using injured patients have focused on imaging injured posterior ligaments.

Other investigations have examined midfoot ligaments in the context of acute sprains or injuries [35,99], and in cases of chronic posterior tibial tendon (PTT) tendinosis [100]. The spring ligament, a critical structure supporting the longitudinal arch in progressive collapsing flatfoot deformity, functions alongside the PTT. Mansour demonstrated significant thickening of the spring ligament in patients with PTT tendinosis, suggesting the compensatory overloading of the ligament [100]. In acute ankle sprains, Allen identified injuries to the spring ligament and/or the talonavicular ligament using ultrasound in conjunction with commonly affected ligaments such as the ATFL [99].

Additionally, ultrasound detected injuries to the fourth and fifth dorsal tarsometatarsal ligaments and the calcaneocuboid ligament in 7.3% and 6.7% of cases, respectively, in the setting of acute ankle sprains [30].

However, none of these studies employed reference standard imaging modalities including the MRI to confirm the ultrasound findings, highlighting a critical limitation in the current body of research.

## 4. Conclusions

MSK ultrasound has been extensively studied for evaluating ligamentous injuries around the ankle, demonstrating consistently high diagnostic accuracy. For the ATFL and CFL injuries, which occur in inversion ankle sprains, diagnostic accuracy is high, with slightly better results in chronic cases, likely due to the feasibility of dynamic assessments, including stress ultrasound. Similarly, in high ankle sprains, ultrasound evaluation of the syndesmosis including the AiTFL has proven highly reliable, with sub-acute cases showing slightly higher accuracy, likely owing to the practicality of both dynamic and static evaluations. For the DL, ultrasound has shown consistently high accuracy in detecting additional DL tears in the context of acute ankle fractures. Future study should focus on exploring the utility of dynamic ultrasound in diagnosing chronic medial ankle instability resulting from chronic DL injury.

## Figures and Tables

**Figure 1 diagnostics-15-00392-f001:**
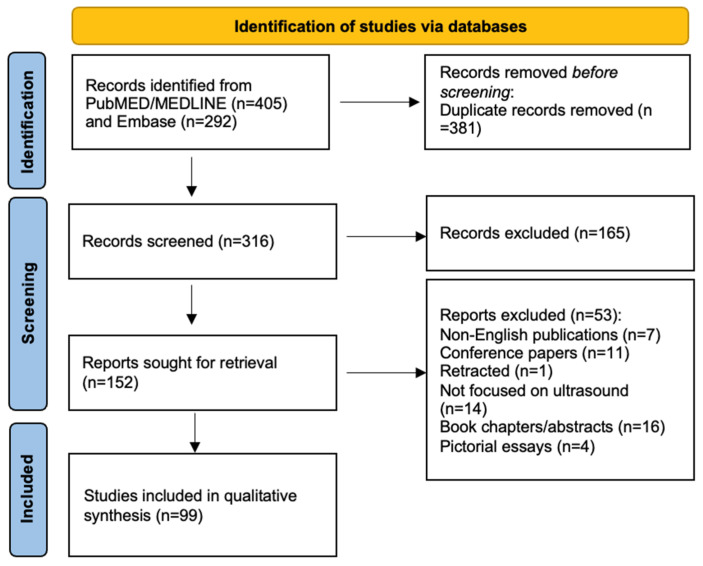
PRISMA Flowchart depicting the literature search methodology.

**Table 3 diagnostics-15-00392-t003:** Clinical studies of the syndesmosis including anterior inferior tibiofibular ligament (AiTFL) ultrasound evaluation above Evidence Level 3.

Author	Year	LOE	Sample Size (Only Syndesmosis if Applicable)	Design	Injury Type	Reference Standard	Key Findings
Heitz [93]	2024	2	26 pediatric patients	Prospective Cohort	Subacute (56 days)	MRI	79% sensitivity, 100% specificity
Ergün [74]	2022	2	30 patients	Prospective diagnostic	Acute	MRI	95.4% sensitivity, 75% specificity
Hosseinian [73]	2022	3	105 patients	Cross Sectional	Acute Chronic	MRI	77% sensitivity and 88% specificity for normal AITFL
Baltes [76]	2021	2	117 patients	Prospective diagnostic	Acute	MRI	100% sensitivity, 100% specificity
Lee [69]	2017	2	85 patients	Prospective diagnostic	Acute on CAI	MRI	100% sensitivity, 100% specificity
Mei-Dan [86]	2009	2	29 patients, 18 controls	Prospective Cohort	Recent injury	MRI	89% sensitivity, 100% specificity
Milz [83]	1998	2	20 patients	Prospective diagnostic	Acute	MRI	66% sensitivity, 91% specificity

CAI, chronic ankle instability.

**Table 4 diagnostics-15-00392-t004:** Clinical studies of deltoid ligament (DL) ultrasound evaluation above Evidence Level 3.

Author	Year	LOE	Sample Size (Only DL if Applicable)	Design	Injury Type	Reference Standard	Key Findings
de Krom [11]	2022	1	12 studies with 890 patients	SR and MA	Ankle fracture	N/A	100% sensitivity, 100% specificity for DL tears
van Leeuwen [13]	2022	1	8 studies	SR and MA	Ankle fracture	N/A	100% sensitivity, 90–100% specificity
Hosseinian [73]	2022	3	105 patients	Cross Sectional	Ankle sprain	MRI	90% sensitivity, 78% specificity for normal DL
Rosa [95]	2020	2	81 patients	Prospective diagnostic	Ankle fracture	GS XR	100% sensitivity, 90% specificity
Kim [96]	2020	2	25 patients	Prospective cohort	Aankle fracture	Arthroscopy	94.7% sensitivity, 66.7% specificity for DDL
Lee [69]	2017	2	85 patients	Prospective diagnostic	Acute sprains on CAI	MRI	100% sensitivity, 95.0 to 97.3% specificity
Henari [97]	2011	2	12 patients	Prospective	Ankle fracture	Arthrogram	100% sensitivity, 100% specificity
Chen [98]	2008	2	17 patients	Prospective observation	Ankle fracture	Surgery and clinical outcome	100% sensitivity, 100% specificity

SR, systematic Review; MA, meta-analysis; DDL, deep deltoid ligament; GS XR, gravitational stress X-ray; CAI, chronic ankle instability.

## Data Availability

Not applicable.

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
