# Peer review of "Role of Ultrasound in Evaluating Ligament Injuries Around the Ankle: A Narrative Review"

_diagnostics, 2025, doi:10.3390/diagnostics15030392_

Round 1

Reviewer 1 Report

Comments and Suggestions for Authors

This review explores recent developments in ultrasound imaging of the lateral, medial, and syndesmotic ligaments of the ankle joint, and discusses its potential applications for evaluating spring and posterior ligaments. This review can assist surgeons in determining the need for ligamentous repair with fracture treatment. Overall, this article is complete, but some issues remain, as described below:

1. Could the authors explain why pictorial essays (or other types of articles) were excluded and give examples of how these articles would affect the reliability and analysis of the results when screened, or the authors could provide relevant literature that would justify this exclusion criterion?

2. The authors mentioned that ‘we focused on healthy human subjects or subjects with suspected or confirmed ligament injuries.’ Could the authors provide more specific details on the inclusion criteria, particularly regarding how ‘suspected’ ligament injuries were defined and diagnosed in the studies included in the review?

3. Furthermore, recent studies have modeled lower extremity ligaments as nonlinear short-term viscoelastic models, which are extremely important in estimating ligament forces: Accurately and effectively predict the ACL force: Utilizing biomechanical landing pattern before and after-fatigue (https://doi.org/10.1016/j.cmpb.2023.107761); New insights optimize landing strategies to reduce lower limb injury risk (https://doi.org/10.34133/cbsystems.0126). These latest relevant studies can enrich the theme of this study, especially in the introduction. It is suggested that the first two paragraphs of the introduction be adjusted.

4. Did the authors consider including other relevant databases, such as Google Scholar, Scopus, or the Cochrane Library, in addition to PubMed and EMBASE? These databases may contain additional literature or systematic reviews that could enhance the comprehensiveness of the search.

5. Many studies mention the sensitivity and specificity of ultrasound in the diagnosis of ankle ligament injuries, but some suggest that MRI may offer higher sensitivity and specificity in some cases. For example, in deep DL injuries, MRI has demonstrated slightly higher diagnostic accuracy. Can you provide more specific comparative data and discuss in detail the complementary nature of ultrasound and MRI in clinical practice?

6. In a review of different literature, some studies used dynamic ultrasound assessment, while others used static ultrasound assessment. Is there evidence that dynamic ultrasound assessment is more effective than static ultrasound assessment in certain types of injuries?

7. For acute injuries of the ATFL and CFL, the article mentions that the diagnostic sensitivity and specificity of ultrasound exhibit significant variability across different studies. Could the authors further discuss the potential factors contributing to these discrepancies? For instance, how might swelling, hematoma, and patient pain in acute injuries affect the accuracy of ultrasound evaluations?

8. The authors discussed different types of ultrasound techniques (e.g., stress ultrasound, dynamic ultrasound, etc.) when evaluating different ligament injuries. Is it possible to further elaborate on the specific advantages and disadvantages of these techniques, especially in different clinical settings (acute injuries, chronic injuries, etc.)?

Reviewer 2 Report

Comments and Suggestions for Authors

The authors review the uses of MSK US in the foot and ankle. This is important since US has many advantages and is likely underutilized in most clinical settings. 

The authors should comment on the underutilization of US, is feasibility a location related issue? Many of the studies were not performed in the United states.. 

Author Response

The authors should comment on the underutilization of US, is feasibility a location related issue? Many of the studies were not performed in the United states.

Thank you for your valuable comment. We completely agree with this comment. MSK ultrasound (MSK US) remains underutilized in the United States compared to its broader adoption in Asian and European countries. Orthopedic surgeons often lack opportunities to learn MSK US due to the absence of dedicated educational programs. I hope this paper contributes to encouraging the integration of MSK US into orthopedic surgery education programs. We added the following paragraph as the 7 th paragraph into the Introduction (Page 3 Line 79-81).  

... This historical progression underscores how MSK ultrasound has transformed from a research tool into a cornerstone of musculoskeletal imaging that is continually evolving to meet the needs of modern clinical practice.

  Despite its diagnostic advantages, MSK ultrasound remains underutilized in clinical practice. This is evident across the world, where limited training during orthopedic residency programs and lack of experienced operators restricts its widespread adoption. Barriers to the adoption of MSK ultrasound include the need for specialized training, limited exposure during medical education, and reliance on more established imaging modalities like MRI and X-ray.

   This paper aims to provide a comprehensive review of the role of MSK ultrasound in diagnosing and treating ligament injuries around the ankle. It focuses on key ligaments, 

Reviewer 3 Report

Comments and Suggestions for Authors

Thank you for the opportunity to review your manuscript, “Ultrasound Evaluation of Ligament Injuries Around the Ankle”.

It is an interesting job. However, there are aspects that I consider should be improved.

1.     First of all, I think the title of the paper should be changed to avoid confusing the reader. I think a correct title could be: "Role of Ultrasound Evaluation of Ligament Injuries Around the Ankle: A Narrative review".

2.     According to the literature, the ATFL is most commonly formed by two fascicles (superior and inferior), while the CFL is a single ligament. The authors do not refer to this issue in their work.

Have you found any publication on this topic that suggests the possibility of ultrasound visualization of the two fascicles of the ATFL?. If the answer is positive, please include it in your article.

Vega, J., Malagelada, F., Manzanares Céspedes, MC. et al. The lateral fibulotalocalcaneal ligament complex: an ankle stabilizing isometric structure. Knee Surg Sports Traumatol Arthrosc 28, 8–17 (2020). https://doi.org/10.1007/s00167-018-5188-8

Author Response

1. First of all, I think the title of the paper should be changed to avoid confusing the reader. I think a correct title could be: "Role of Ultrasound Evaluation of Ligament Injuries Around the Ankle: A Narrative review".

Thank you for your valuable suggestion. We totally agree with this comment. We changed the title as per your comment into "Role of Ultrasound in Evaluating Ligament Injuries Around the Ankle: A Narrative Review"

2.  According to the literature, the ATFL is most commonly formed by two fascicles (superior and inferior), while the CFL is a single ligament. The authors do not refer to this issue in their work. Have you found any publication on this topic that suggests the possibility of ultrasound visualization of the two fascicles of the ATFL?. If the answer is positive, please include it in your article.

Thank you for your valuable comment. In response, we identified a paper by Kakegawa et al. that demonstrated the visualization of the two fascicles of the ATFL using ultrasound. While this study is conducted in a laboratory setting rather than a clinical one, we have included this reference in our manuscript as it aligns with the context of our discussion. We sincerely appreciate your insightful suggestion.

Kakegawa et al. Relationship between inferior fascicle of anterior talofibular ligament and articular capsule in lateral ankle ligament complex. Surg Radiol Anat. 2022 Feb;44(2):253-259. doi: 10.1007/s00276-021-02851-1.

Round 2

Reviewer 1 Report

Comments and Suggestions for Authors

Thanks for the response from the authors. The quality of the article has already improved a lot after the revisions. I have only a few minor questions and suggest that it be accepted with revisions. The current study focuses on the foot and ankle joints such as the ATFL and CFL, yet there is a lack of relevant descriptions of ankle injuries and their importance in the lower extremity dynamic chain. Therefore, it is recommended that the authors add the description related to the importance of the ankle joint and lateral ligament injuries at the beginning of the third paragraph in the introduction. The application of MSK ultrasound in the diagnosis of ankle injuries was then taken up, and it is believed that this will better refine the logic and content. For the most recent literature on ankle-related studies, authors may consider referring to the previously mentioned studies (previous comment 3).

Author Response

The current study focuses on the foot and ankle joints such as the ATFL and CFL, yet there is a lack of relevant descriptions of ankle injuries and their importance in the lower extremity dynamic chain. Therefore, it is recommended that the authors add the description related to the importance of the ankle joint and lateral ligament injuries at the beginning of the third paragraph in the introduction. The application of MSK ultrasound in the diagnosis of ankle injuries was then taken up, and it is believed that this will better refine the logic and content. For the most recent literature on ankle-related studies, authors may consider referring to the previously mentioned studies (previous comment 3).

Thank you for your insightful comment. We fully agree with your observation and have revised the manuscript accordingly. To emphasize the significance of ankle joint injuries, we added the following statement at the beginning of the third paragraph in the introduction (Line 45-47):

“Ankle injuries encompass both sprains and fractures. Ankle sprains are among the most common sports-related injuries, while ankle fractures rank as one of the most frequent injuries to the lower extremity.”
